# The Effect of Steviol Glycosides on Sensory Properties and Acceptability of Ice Cream

**DOI:** 10.3390/foods11121745

**Published:** 2022-06-14

**Authors:** Nannapas Muenprasitivej, Ran Tao, Sarah Jeanne Nardone, Sungeun Cho

**Affiliations:** 1Department of Poultry Science, Auburn University, Auburn, AL 36849, USA; nzm0064@auburn.edu (N.M.); sjn010@auburn.edu (S.J.N.); 2Department of Food Science & Human Nutrition, Michigan State University, East Lansing, MI 48824, USA; taor.0908@gmail.com

**Keywords:** steviol glycosides, rebaudiosides, no sugar added, aftertaste, minor steviol glycosides

## Abstract

There has been a challenge in overcoming the bitter aftertaste of stevia, a natural non-caloric sweetener. Recent research focuses on investigating various types of steviol glycosides, the sweet compounds in stevia leaves, as they exhibit different sensory characteristics. This study determined the sensory properties and acceptability of ice cream sweetened solely with three steviol glycosides, rebaudioside (Reb) A, D, and M (0.09% *w*/*v*), using sucrose-sweetened ice cream as a control (14% *w*/*v*). Ice cream consumers (*n* = 92) rated their overall liking, attribute liking, and sweetness and bitterness intensities and described the aftertastes of each sample using check-all-that-apply. The liking scores of Reb D- and M-sweetened ice creams were significantly higher than those of Reb A-sweetened ice cream. Among the three glycosides, only Reb M showed a sweetness intensity comparable with that of sucrose. Consumers perceived the aftertastes of Reb D and M ice creams as being more *sweet*, *pleasant*, *creamy*, and *milky*, while Reb A was more *artificial* and *chemical*. Reb D and M ice creams were also plotted close to sucrose in the correspondence analysis graph, meaning that their aftertaste characteristics were similar to those of sucrose. The present study clearly highlights that Reb D and M have better tastes and provide better perceptions to consumers than Reb A, which is the most widely used glycoside in food industry.

## 1. Introduction

According to the American Heart Association (AHA), the daily recommended consumption of sugar should be no more than 36 g for men and no more than 25 g for women. The average American adult consumes twice the daily amount of recommended sugar [1,2], leading to the development of chronic diseases and diabetes in consumers [1,3,4]. This has resulted in food industries investing in finding alternatives to sucrose in the form of high-intensity sweeteners (HISs). HISs approved by the U.S. Food and Drug Administration (FDA) include six artificial sweeteners (saccharin, aspartame, acesulfame potassium (Ace-K), sucralose, neotame, and aspartame) and two natural sweeteners—stevia (steviol glycosides) and monk fruit (luo han guo fruit extract) [5]. These HISs are used in very small amounts while providing low to almost negligible caloric content to food and beverages [6]. However, the effects of consuming artificial HISs on health and metabolism are not well established [7,8]. Some studies have found no adverse effect on diabetics’ blood glucose after they consumed artificial sweeteners [9]. Others claimed that the consumption of artificial sweeteners could trigger a small rise in insulin levels [10], negatively change gut bacteria [11], and could have a positive association with obesity [11,12,13]. These contradictory results have negatively impacted consumers’ perceptions of artificial HISs [14] and increased their interest in natural sweetener options such as stevia [15]. Not only is stevia a natural non-caloric HIS, it also lowers the glycemic index (GI) [16] and blood glucose levels [16]. In addition, Anton et al. [17] discovered that the consumption of stevia significantly helped to reduce the food intake of consumers as compared to sucrose (*p* < 0.01).

Many food companies have started to introduce stevia into their food and beverage products to promote healthy options to consumers [1,18,19]. Stevia has been widely used in beverage drinks more commonly than other HISs (e.g., aspartame) [20] because of its advantages over sucrose and artificial sweeteners. According to Statista research department [21], the global market value of stevia was predicted to reach over USD 770 million by 2022. Furthermore, according to Allied Market Research in 2020, the global market of stevia is also predicted to reach about USD 1.2 billion by 2026, with a compound annual growth rate (CAGR) of eight percent from 2019 to 2026 [22]. Moreover, natural HIS options, including stevia, have been utilized in dairy products as it grows in popularity among consumers and food industries. For instance, ice cream, a popular dessert in the U.S. [23], typically contains an average of 15% sucrose [24]. The demand for naturally sweetened ice cream products has become a challenge to the industry, leading companies to launch healthy options for consumers, such as low-fat, low-sucrose, or no-sucrose-added products [25,26]. Currently, brands such as Halo-Top [27] and Enlightened [28] utilize either stevia extract (a mixture of steviol glycosides) or pure Reb A with other caloric or non-caloric sweeteners. However, none of the companies use stevia as a sole sweetener.

Stevia (*Stevia rebaudiana*) is a South American plant which has up to 300 times the sweetness of sucrose [29,30]. The natural constituents of the stevia leaves, steviol glycosides, have been generally recognized and considered as safe (GRAS) in the U.S. since 2008 [5,31]. Leaves of stevia contain sweet compounds are known as either diterpene glycosides or (more commonly) steviol glycosides [32,33], of which more than 40 steviol glycosides identified [33,34,35,36]. Most of them lack relevant sweetness data, except for the following 11 types: stevioside, rebaudioside A (Reb A), Reb B to F [37,38], M [32,39], steviolbioside [39], rubusoside [32,35], and dulcoside A [32,35]. Moreover, these steviol glycosides are currently approved in the European markets [40,41]. The major types of steviol glycosides are stevioside (110–270 times sweeter than sucrose) [32,35] and Reb A (250–400 times sweeter than sucrose) [32,36,37] which are found at about 4–13% and 2–4% in driedstevia leaf, respectively [29,30,37]. However, these major steviol glycosides have been found to provide a significant bitter aftertaste [33,36,41]. Thus, many researchers have investigated various steviol glycosides [39,42] and found that certain minor types of steviol glycosides, such as Reb D and M, exhibited different levels of sweetness and bitterness from the other major types, but are found only 0.1–0.2% [30]. Reb M displays fast sweetness onset, reduces the non-sweet taste, and results in less lingering bitterness when compared to Reb A [39]. Similarly, using trained panelists, it was found that Reb D elicits significantly less bitterness than Reb A at similar levels of sweetness [43]. In addition, using a consumer panel, Tao and Cho [44] found that both Reb D and M showed better taste characteristics than Reb A (e.g., less bitterness) in a water solution at 0.09% (*w*/*v*). They also found that the aftertaste descriptors of Reb D and M were close to sucrose. However, no study to date has investigated the sensory properties of Reb D and M compared to Reb A in food matrices.

In this study, we investigated the sensory characteristics and acceptability of ice cream when sweetened solely with Reb A, D, and M. Using a consumer panel, consumer perceptions of ice cream sweetened with the minor glycosides (Reb D and M) and the major glycoside (Reb A) were compared to sucrose-sweetened controls. Furthermore, we investigated whether Reb A, D, and/or M could be used as sole sweeteners in high-sucrose applications, without compromising sensory quality.

## 2. Materials and Methods

### 2.1. Materials

The materials used to produce ice cream samples were purchased from a local grocery store: heavy cream (Horizon Organic, Broomfield, CO, USA), non-fat dry milk (Kroger, Cinciannati, OH, USA), vanilla extract (Spice Island, B&G Foods Inc, Parsippany-Troy Hills, NJ, USA), polydextrose (Litesse, DuPont, Wilmington, DE, USA), and sucrose (Smidge & SpoonTM, Kroger, Cincinnati, OH, USA). The steviol glycosides used in the ice cream for the study were high purity (95%) Reb A, Reb D, and Reb M from Sweegen (Santa Margarita, CA, USA).

### 2.2. Ice Cream Preparation

Reb A, D, and M were used at 0.09% (*w*/*v*) in the ice cream formulation. The 0.09% concentration was chosen because it was found to have a similar sweetness level as the 14% sucrose (*w*/*v*) from the previous study by Tao and Cho [44], which is within the sweetness level range for frozen desserts and ice cream. Table 1 shows ice cream formulations and indicates the functionality of each ingredient used in this study. The dry ingredients (non-fat dry milk, polydextrose, and sucrose or Reb A, D, or M) were first blended in the mixer (KitchenAid, St. Joseph, MI, USA) until they were homogenized, followed by the addition of warm water (~43 °C). Next, heavy cream and vanilla extract were added with continuous stirring until the mixture was homogenized. The ice cream mixture was aged for one hour at 4 °C and then place in the ice cream maker for one hour (Cuisinart, Stamford, CT, USA). The ice cream was transferred into a plastic container (64 oz) and stored in a walk-in freezer at −20 °C. Table 1 also shows the caloric values of each ice cream, which was based on 80 g or 2/3 cups (i.e., ice cream serving size). It was generated using Genesis R&D Supplement Formulation & Labeling Software (ESHA Research, Oak Brook, IL, USA).

### 2.3. Panel Recruitment

Consumer panelists who consume ice cream (at least 2–3 times per month) and zero-calorie sweeteners (at least once a month) were recruited from Auburn University (18–65+ years old). The pre-survey was performed using Qualtrics online survey software (Qualtrics, LLC, Provo, UT, USA), including the consumption behavior of HISs and the frequency of ice cream consumption.

### 2.4. Sample Preparation

All ice cream samples were made two days before the test. One day before the test, a scoop (~30 g) of ice cream was transferred into a 2 oz plastic soufflé cup labeled with random 3-digit coded numbers. They were stored in a walk-in freezer (−20 °C).

### 2.5. Testing Procedure

This study was approved by the University Institutional Review Board of Auburn University (Auburn, AL, USA) (Protocol #: 21-204 EX 2104). RedJade sensory science software (RedJade Sensory Solutions LLC, Redwood City, CA, USA) was used to collect data during the entire testing.

After confirming that the code of the served sample matched the code on the screen, the panelists were asked to taste a spoonful of the ice cream sample (less than 1/2 of the serving cup) to evaluate their overall liking and attribute liking (appearance, flavor, texture/mouthfeel, and aftertaste) of the sample using a 9-point hedonic scale (1 = Dislike extremely, 9 = Like extremely). Next, they were instructed to take another spoon of the same sample and swallow it to evaluate the aftertaste of each sample. For the aftertaste, they were asked to rate the intensities of sweetness and bitterness using a 15 cm-line scale (0 = Not at all sweet/bitter, 15 = Extremely sweet/bitter) and then to choose aftertaste descriptors using check-that-all-apply (CATA) analysis. The listed attributes for CATA included *artificial*, *metallic*, *milky*, *buttery*, *chemical*, *bitter*, *spicy*, *vanilla*, *honey*, *minty*, *pleasant*, *tart*, *sweet*, *and spicy*. This study used 11 terms from a previous study conducted by Tao and Cho [44], who evaluated the aftertaste of stevia solutions. The terms *buttery*, *creamy*, and *milky* were added to describe the flavor attributes of ice cream. Lastly, the term *spicy* was used as an attention check. The purchase intent question, using a 5-point Likert scale (1 = Definitely would not buy, 5 = Definitely would buy), was asked at the end of each sample. A 30 s break was enforced before receiving the next sample. During the break, water and unsalted crackers were provided as palate cleansers. After evaluating all four samples, consumer behavior and demographic questions were asked, including low/no-sugar product consumption behavior, familiarity with zero-calorie sweeteners (i.e., aspartame, ace-k, erythritol, monk fruit, saccharin, stevia, sucralose, and xylitol), health-related questions (health conditions of the panelists and their family and diet), and demographic questions (i.e., age, gender, height, weight, education level, ethnicity, and household income).

### 2.6. Statistical Analysis

Data analysis was performed using XLSTAT (AddinSoft, New York, NY, USA). The sensory evaluation questions and the sweetness and bitterness intensities were determined by two-way analysis of variance (ANOVA), with a 95% confidence level (*p* < 0.05), and Tukey’s (HSD) tests, treating ice cream samples as a fixed effect and the consumer panel as a random effect. To determine if there was an interaction effect between the overall liking score of each ice cream and gender, data were also analyzed using two-way ANOVA with one interaction effect (fixed effects: ice cream sample and gender). Cohran’s Q test was used to analyze the check-all-that-apply (CATA) responses to determine any significant differences between ice cream samples. Correspondence analysis (CA) was used to show the relationship between sensory attributes and the samples.

## 3. Results

### 3.1. Participants’ Characteristics

A total of 92 participants who consumed ice cream at least two to three times per month completed the study. The age range was between 18 and 65 years old, with an average body mass index (BMI) of 26.0 ± 5.3 kg/m^2^. Table 2 shows the socioeconomic statuses of the panelists. We recruited female and male participants (59.8% and 40.2%, respectively). The majority variables of the panel were consumers aged 18–25 years old (51.1%), with the highest education level being the graduate degree (38.0%). Therefore, most panelists received household incomes under USD 30,000 (72.8%). Lastly, the majority of participants were White or Caucasian (70.6%).

Table 3 shows the ice cream consumption behaviors of the panelists. Over 80% of the consumer panel consumed ice cream at least once a week, and more than 90% of them purchased ice cream at least once a month. However, only 31.5% of the ice cream consumers purchased low- or no-sugar ice cream within the past six months.

Table 4 shows how many different types of sweeteners the consumer panel could recognize. They were required to select each sweetener from ‘Very unfamiliar’ to ‘Very familiar’. Among all-natural sweeteners, stevia was picked the most for ‘Very familiar’, while monk fruit was picked the most for ‘Very unfamiliar’ (41.3% and 4.3%, respectively). For the artificial sweeteners, the consumer panel frequently selected ‘Very familiar’ for sucralose (33.7%) and ‘Very unfamiliar’ for Ace-K (72.8%).

Table 5 shows consumption behaviors of low/zero-sugar products and various sweeteners by the consumer panel. A total of 52 (56.6%) participants consumed low/zero-sugar foods and/or beverages at least once a month. Out of 92 participants, only 28.3% consumed stevia at least once a month. This shows that they were not frequent stevia users, although the majority of participants were very familiar with stevia among all artificial and natural sweeteners. 

### 3.2. Sensory Analysis of Ice Cream

Table 6 summarizes the means ± standard error (SE) of overall liking, attribute liking, and purchase intent of each ice cream sample evaluated by the ice cream consumers (*n* = 92).

Sucrose ice cream received scores of over 7.0 (moderately like) on a nine-point hedonic scale and was significantly higher than all three steviol glycosides for overall liking (*p* < 0.05) and every attribute liking (*p* < 0.001) except for appearance and texture/mouthfeel liking. There were no differences between minor steviol glycosides (Reb D and M) and sucrose ice cream on the appearance liking (*p* = 0.063 and *p* = 0.183, respectively). In the texture/mouthfeel liking, Reb M and sucrose ice cream were found not to be significantly different from one another (*p* = 0.052), but the *p*-value was close to the significance level (*p* < 0.05). Although Reb M ice cream received slightly higher liking scores than Reb D ice cream in every category, they were not significantly different (Table 6). There were significant differences in hedonic impressions between the major steviol glycoside (Reb A) and the minor steviol glycosides (Reb D or Reb M). Reb A ice cream was liked significantly less than Reb D and M in every category except for texture/mouthfeel and appearance liking. All steviol glycoside ice creams showed similar scores in appearance liking, with the range of 6.9 ± 0.67. For purchase intent, sucrose ice cream received a score of 3.7 on a five-point Likert scale, which was close to four ‘Probably buy’. The purchase intent of Reb A was close to two ‘Probably would not purchase’ (i.e., 2.1). Reb D and M ice cream samples were rated significantly higher than Reb A (*p* < 0.001), both of which were close to three ‘Might or might not purchase’ (i.e., 2.6 and 2.8, respectively). Reb M ice cream received the highest purchase intent score among the steviol glycoside ice creams, although there was no significant difference between Reb D and M (*p* = 0.49).

Next, the participants were asked to put each sample on their tongues for 5 s and then swallow the sample. Immediately after swallowing, they were asked to determine the intensities of sweetness and bitterness using a 15-cm intensity line scale. Table 7 shows the means (±SE) of the sweetness and bitterness intensities of the ice cream samples.

Among all three steviol glycosides, Reb M ice cream received the highest sweetness intensity and showed comparable sweetness to that of sucrose ice cream (*p* = 0.609). Reb A and D ice cream samples were both significantly less sweet than sucrose and Reb M ice cream (*p* < 0.001). Moreover, Reb M and sucrose ice cream samples received bitterness intensity ratings that showed no significant difference (*p* = 0.175), while the intensity score of Reb M ice cream was higher than that of sucrose ice cream (i.e., 2.6 and 1.6 on a 15-cm line scale, respectively). Reb A ice cream received the highest bitterness score among all the samples (*p* < 0.001). There was no significant difference in the bitterness intensity between Reb D and M ice cream (*p* = 0.853). The participants rated the bitterness intensity of Reb D and M at 2.9 and 2.6 on a 15-cm line scale, respectively, while Reb A received 5.4.

Table 8 shows the aftertaste attributes for each ice cream sample as selected by the consumer panelists (*n* = 92).

Reb D and M ice cream samples received no significant difference from each other for each aftertaste term (Table 8). The terms *bitter*, *metallic*, *milky*, and *tart* were used to describe Reb D, Reb M, and sucrose ice creams. Interestingly, for the term *artificial*, Reb D ice cream was chosen by 12 more panelists than Reb M (47 vs. 35 for Reb D and M, respectively), but there was no significant difference between them. However, the term *artificial* was used to describe Reb A significantly more than Reb M (54 vs. 35, respectively) (*p* < 0.0001), but there was also no significant difference between Reb A and Reb D (54 vs. 47, respectively). The results for the term *sweet* across all ice cream samples was complementary to a 15-cm line intensity scale in that the sucrose and Reb M ice creams were chosen by most panelists (72 vs. 62, respectively) and Reb A and D ice creams were chosen the least frequent with this term (39 vs. 53, respectively). The term *bitter* was chosen the most with Reb A ice cream by a consumer panel (37), which was similar to the bitterness intensity scale (Table 7). However, for the term *bitter*, Reb D (14) was described similarly to both sucrose and Reb M ice creams (2 and 10, respectively). Reb A ice cream was described more frequently with negative terms than other ice cream samples, including *bitter*, *metallic*, and *tart*. However, all three steviol glycosides shared *metallic*, *vanilla*, and *tart* terms with no significant difference (*p* < 0.05). Three terms which were not significantly different from each other were *buttery*, *honey*, *and minty*.

The sensory attributes of the sweeteners were summarized visually in Figure 1. The first two dimensions explained 96.96% of the variation. The terms *pleasant*, *vanilla*, *sweet*, and *creamy* were associated with and chosen for sucrose more than for all three steviol glycosides. Reb A was associated to more negative terms, including *metallic*, *bitter*, *chemical*, and *tart*, while Reb D and M were mostly associated with positive words. Moreover, both minor steviol glycosides were plotted close to each other and were closer to sucrose when compared to Reb A.

In this study, it was later found that male (*n* = 37) and female panelists (*n* = 55) rated their overall liking of the ice creams differently from each other. Figure 2 indicates that male panelists gave lower mean overall liking scores for all steviol glycoside ice creams (i.e., Reb A, D, and M) than for sucrose ice cream (5.6, 6.2, 6.2 vs. 7.7, respectively). On the other hand, female panelists gave both sucrose and Reb M ice creams a similar liking score of seven points (7.6 and 7.0, respectively), followed by Reb D (6.5) and Reb A (5.3).

## 4. Discussion

Sensory Evaluation

This study examined the sensory quality of three different steviol glycosides (0.09% *w*/*v*) in ice cream and compared them against sucrose ice cream (14% *w*/*v*) as a control. Stevia has different physiochemical properties than sucrose [16], one of which is that it lacks a bulking agent. Thus, this negatively affects the texture of stevia ice cream samples. Therefore, we used polydextrose as a bulking agent when using stevia instead of sucrose in ice cream formulations (Table 1). Polydextrose has a variety of functional properties with potential health benefits, making it a great additive in various food products [47]. Not only does polydextrose aid in enhancing ice cream texture, it also acts as a fat replacer to improve the appearance and the mouthfeel of the ice cream [48]. This allows the ice cream made with stevia to acquire some sensory characteristics similar to those of sucrose (i.e., appearance and texture/mouthfeel attributes). However, polydextrose contains 1 Kcal per gram, which adds additional calories to stevia ice cream samples. Despite small differences in caloric intake between sucrose and stevia ice cream (150 vs. 120 Kcal, Table 1), this ice cream formulation with stevia is suitable for people with diabetes who are looking for an ice cream option with no sugar while having a similar texture/mouthfeel as regular ice cream. Polydextrose, similarly to stevia, does not affect blood glucose levels [45,49]. Additionally, the human body does not metabolize stevia, meaning we obtain no calories from consumption [16,40,46].

The results from Table 6 show significant differences in overall liking and attribute liking scores among stevia ice cream samples (Reb A, D, and M) at 0.09% (*w*/*v*) and sucrose ice cream samples at 14% (*w*/*v*). When comparing Reb D and M (the minor steviol glycosides) with Reb A (the most widely used steviol glycoside in the food industry), Reb A ice cream was least preferred by panelists; it was given the lowest score among samples in all hedonic liking scores. The consumer panel preferred minor steviol glycosides over the major steviol glycoside, except in appearance and texture/mouthfeel attributes (*p* < 0.001). Reb D and Reb M ice creams scored at around six points (Like slightly) in all hedonic liking scores except aftertaste liking, which were both at around five points (Neither like nor dislike). They both shared similar scores and showed no significant difference, but the consumer panel showed a slightly higher preference for Reb M over Reb D ice cream. Moreover, Reb M and sucrose ice creams shared more similar attributes than other steviol glycosides (i.e., appearance and texture/mouthfeel). According to Everitt [50], a mean liking score of seven or higher on a nine-point hedonic scale is acceptable for sensory quality. Even though the replacement of sucrose by high-intensity sweeteners can negatively alter the perception of bitter and sweet taste [51], we found that these minor steviol glycosides were nearly as good as sucrose ice cream. The mean purchase intent was scored the least with Reb A ice cream, followed by Reb D and M ice creams, then sucrose ice cream. This pattern was reflected in overall liking and attribute liking scores (Table 6). The nine-point hedonic scores and five-point Likert score from this study confirmed the better effect of utilizing minor steviol glycosides (especially Reb M) as sucrose substitutes, rather than Reb A, in food matrices.

While many studies have developed ice cream formulations with different ratios of stevia and other sweeteners, few studies have incorporated formulations using purely stevia. Alizadeh et al. [16] used five different ratios of sucrose and stevia in ice cream and compared them against the control (sucrose only) using a five-point intensity score (zero = uncharacterized intensity and five = very strong intensity). One of the ratios, 9.3 g sucrose and 0.04 g stevia, was found to receive high liking scores in taste, texture, and overall liking, among four other different ratios. However, the control still maintained the highest liking scores for flavor, taste, and mean liking scores. The authors found that the substitution of sucrose with stevia negatively affected the liking scores of panelists [16]. This assumes that panelists do not prefer the product with stevia. To address this point, Alizadeh et al. [16] used nearly pure steviol glycosides to test consumer acceptability in ice cream, using a purification rate of 90%. McCollum [52] claimed that a high percentage of purity of stevia indicated purer extraction, which brings a sweeter taste and hinders the bitter aftertaste of steviol glycosides. In another study, Velotto et al. [53] used solely >98% Reb A stevia extract powder and compared it against sucrose (control) at 26.1% in both traditional (1.0% Reb A) and vegan ice cream (1.5% Reb A) samples. The results showed that both traditional and vegan ice cream sweetened with stevia received significantly higher scores than sucrose samples in sweet taste/flavor and overall taste attributes (*p* < 0.05). Thus, a high-purity stevia extraction method could mitigate negative aftertastes (i.e., bitter and lingering) found in stevia, especially Reb A at a high concentration. This could be the potential reason explaining the consumer panel’s preference for minor steviol glycoside ice cream over Reb A ice cream.

On the sweetness and bitterness intensity scales (Table 7), the consumer panel gave Reb M and sucrose ice creams similar scores, with no significant difference (*p* = 0.220 and *p =* 0.175, respectively). This could explain why Reb M received the highest hedonic and Likert scores among all steviol glycoside samples, even though Reb D and M ice cream samples received bitterness intensity scores of 2.9 and 2.6, with no significant difference (*p* < 0.05). A study done by Jung et al. [54] found that Reb D (0.0209%) and Reb M (0.0190%) exhibited a similar bitterness aftertaste, with no significant difference (*p* < 0.05). In addition, Tao and Cho [44] found that Reb D and M solutions were not significantly different in terms of in-mouth and lingering sweetness (*p* = 0.05) at the same concentrations as this study. However, their consumer panelists were able to distinguish Reb D from Reb M because the Reb M solution provided the highest immediate sweet taste among other samples. Thus, in this study, Reb M ice cream was found to be sweeter than either Reb A or D ice creams. The sweet taste attribute of steviol glycosides is dependent on the functional group (R-groups) at positions C-13 and C-19 of the steviol core, on which different types of sweet molecules are attached [32,36]. The main difference between Reb A and Reb D and Reb M is the number of glucose molecules positioned at the C-19 [36]. Reb A only has one glucose moiety, while Reb D has two and Reb M has three. This makes both minor steviol glycosides provide a sweeter taste and a less bitter aftertaste than Reb A [36,42,55]. This finding corresponds with the bitterness intensity score being highest for Reb A among the three steviol glycoside ice cream samples (Table 7). The chemical compounds of the different steviol glycosides may affect the taste in both solutions [36,39] and food matrices encountered by the consumer panel. 

In the CATA analysis (Table 8), consumer panelists selected every aftertaste term for Reb D and M ice creams with similar frequencies to one another. They were described with several positive attributes such as *milky*, *vanilla*, and *pleasant*, and they were also plotted close to sucrose in the corresponding analysis. Reb A ice cream was on the opposite side of sucrose ice cream, with more negative attributes such as *bitter* and *metallic* (Figure 1). All steviol glycoside ice cream samples were described with the term *metallic* and *tart*, although Reb D and M were chosen less frequently than Reb A. However, Tao and Cho [44] found that there was no significant difference with the term *metallic* between all steviol glycosides (Reb A, D, and M) and the sucrose solution. In this study, the term *artificial* was selected to describe all three steviol glycoside ice creams more than sucrose ice cream but was used to describe Reb M ice cream significantly less than Reb A and D ice creams (*p* < 0.0001). On the other hand, Tao and Cho [44] found that there was no significant difference between all steviol glycoside solutions (Reb A, D and M) at the same concentration, and the sucrose solution was least frequently chosen among the samples (*p* <0.001). 

Although steviol glycoside extracts are known for their sweetness, many elicit undesirable aftertastes, including bitterness [43]. In this study, we confirmed that these minor glycosides give a significantly less bitter aftertaste than Reb A in food matrices, especially at high-sucrose concentrations. The result for Reb A ice cream from CATA with the term *bitter* corresponded to the bitterness intensity rating on a 15-cm line scale (Table 7 and Table 8). Many research and food industries have been investigating the sensory analysis of the minor steviol glycosides because they provide more sweet and less bitter taste profiles [39,56]. Although we observed that these minor steviol glycosides contained some negative terms, they were more positively associated with sucrose ice cream than Reb A ice cream.

Our CATA analysis revealed that consumer panelists associated Reb M and sucrose with the term *sweet* with similar frequency. Although Reb D and M shared similar scores in every aftertaste descriptor, the consumer panel selected the *sweet* attribute with similar frequency for both sucrose and Reb M ice creams (72 and 62, respectively). This supports previous findings suggesting that Reb M has the highest sweetness intensity compared to other steviol glycosides [39,42,44].

There was no interaction between male and female preferences for ice creams sweetened with steviol glycosides versus sucrose (*n* = 37 and *n* = 55, respectively). The *p*-value of the interaction effect was 0.06, which is slightly greater than 0.05, the significance level, even though the *p*-value could be changed with the unequal sample size of the panelists for each gender. However, it is important to note that female participants clearly showed a higher preference for the Reb D and M ice creams than male participants did (Figure 2). This may be because sugar-free products are more popular among female participants than males and as such they are more familiar with the taste of zero-calorie sweeteners. Several studies have shown that women tend to choose healthier food choices than men [57,58], making them a targeted consumer for ice cream sweetened with stevia. Therefore, it would be interesting to further investigate the gender differences in terms of preferences and perceptions of stevia-sweetened products.

One possible limitation of this study could be the participants’ household incomes. There are many factors that affect the consumer’s purchase intent, one of which is the price [59]. The majority of participants (~70%) earned an income of less than $30,000 because a majority of participants were still in college (four-years college and graduate degree). Consumers were asked if they were willing to purchase ice cream samples for $4.99 per pint using a five-point Likert scale. Therefore, the skew of this demographic might affect the purchase intent score. Another limitation of this study could be that we only used a pure vanilla flavor in this study. If strawberry or chocolate flavors were used in the ice cream, the stronger flavors might be able to mask the unpleasant aftertastes of stevia and might increase liking scores.

## 5. Conclusions

This study has confirmed that minor steviol glycosides, Reb D and M, had positive effects on the acceptability of zero-sugar ice cream when compared with Reb A, the major steviol glycoside. Although sucrose ice cream received the highest liking scores among ice cream samples, these minor steviol glycosides received overall liking scores between ‘like slightly’ and ‘like moderately’, which were significantly higher than those of Reb A. Furthermore, the aftertaste characteristics of Reb D and M were comparable to sucrose ice cream. Interestingly, only Reb M was found to provide a sweetness taste profile similar to sucrose, but there was no significant difference in flavor liking of Reb D and M ice creams. Thus, it is suggested that Reb D and M can be used as natural non-caloric sweetener options to replace sucrose without adding a bitter aftertaste, even in high-sugar applications such as ice cream or frozen desserts. However, further studies are needed to determine the commercial applications for Reb D and M, as their quantities in stevia leaves are small. Breeding stevia plants for increased concentrations of these minor glycosides would be beneficial for increasing the supply of the desired glycosides. It would also be beneficial to identify an optimal combination that uses both major and minor steviol glycosides to accommodate the bitter aftertaste issue from Reb A and the small extractable quantities of Reb D and M.

## Figures and Tables

**Figure 1 foods-11-01745-f001:**
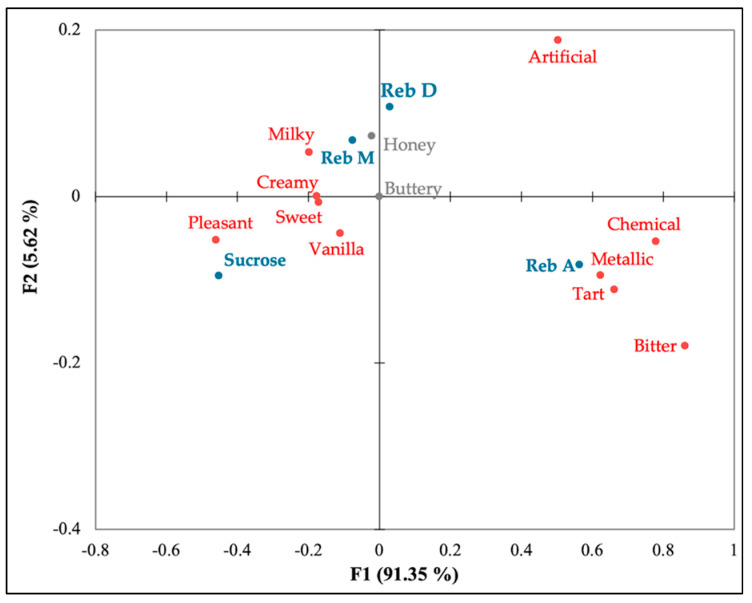
Correspondence analysis (CA) of each ice cream. Blue indicates samples; red indicates significant attributes; grey indicates not−significant attributes.

**Figure 2 foods-11-01745-f002:**
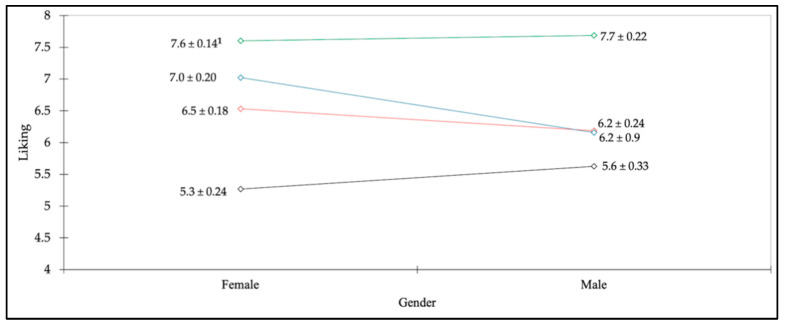
^1^ The mean overall liking scores (±standard error) of each ice cream between female and male participants. Green indicates sucrose ice cream; blue indicates Reb M ice cream; red indicates Reb D ice cream; black indicates Reb A ice cream.

**Table 1 foods-11-01745-t001:** The functionality of each ingredient for ice cream formulation and caloric values.

Ingredients	Functionality	Sucrose (g)	Stevia (Reb A, D, M) (g)
Heavy cream	Mouthfeel texture [45]	400.0	400.0
Non-fat dry milk	Texture and flavor [46]	140.0	140.0
Water	Solvent [24]	650.0	650.0
Vanilla extract	Flavoring agent	5.0	5.0
Sucrose	Sweetener	203.0	0.0
Reb A, D, M	Sweetener	0.0	1.3
Polydextrose	Bulking agent [45]	50.0	245.0
Total		1448.0	1441.3
Calories per serving ^1^ (80.0 g or 2/3 cup)		150.0	120.0

^1^ The caloric values were generated by Genesis R&D Supplement Formulation & Labeling Software (ESHA Research, Oak Brook, IL, USA).

**Table 2 foods-11-01745-t002:** Demographic and socioeconomic characteristics of the consumer panelists (*n* = 92).

Variable	Definition	Participant (*n*)	Frequency (%)
Gender			
	Female	55	59.8
	Male	37	40.2
Age			
	56–65	4	4.4
	46–55	3	3.3
	36–45	5	5.4
	26–35	33	35.9
	18–25	47	51.1
BMI (Mean ± Standard Deviation)		26.0 ± 5.3 kg/m^2^	
Education level			
	Graduate degree (Master’s, Doctorate, etc.)	35	38.0
	4-year college degree	28	30.4
	2-year college degree	5	5.4
	High School diploma or GED	24	26.1
Household income			
	Over $80,000	4	4.4
	$50,000 to $79,999	7	7.6
	$30,000 to $49,999	14	15.2
	Under $30,000	67	72.8
Ethnicity			
	Asian or Pacific Islander	11	12.0
	Black or African American	2	2.2
	Hispanic or Latino	13	14.1
	White or Caucasian	65	70.6
	Prefer not to say	1	1.1

**Table 3 foods-11-01745-t003:** Ice cream consumption behaviors by the consumer panel (*n* = 92).

Variable	Definition	Participants (*n*)	Frequency (%)
Frequency of ice cream consumption			
	2–3 times per month	16	17.4
	Once a week	33	35.9
	2–3 times per week	37	40.2
	More than 3 times per week	6	6.5
Frequency of ice cream purchase			
	Once every 2 or 3 months	6	6.5
	Once a month/every four weeks	17	18.5
	Once every 2 or 3 weeks	50	54.4
	Once a week or more often	19	20.7
Low- or no-sugar ice cream purchased within the past six months			
	Yes	28	30.4
	No	59	64.2
	Don’t remember	5	5.4

**Table 4 foods-11-01745-t004:** Familiarity of various sweeteners selected by consumer panel (*n* = 92).

	Familiarity, *n* (%)				
Low/Zero-Sugar Sweeteners	Very Unfamiliar	Somewhat Unfamiliar	Neutral	Somewhat Familiar	Very Familiar
Artificial sweeteners					
Acesulfame-K	67(72.8%)	14(16.3%)	4(4.3%)	3(3.3%)	4(4.3%)
Aspartame	25(28.3%)	9 (9.8%)	3(3.3%)	29(31.5%)	26(28.3%)
Erythritol	62(68.5%)	10(10.9%)	5(5.4%)	10(10.9%)	5(5.4%)
Saccharin	22(26.1%)	7(7.6%)	10(10.9%)	26(28.3%)	27(30.4%)
Sucralose	17(18.5%)	7(7.6%)	5(5.4%)	32(34.8%)	31(33.7%)
Natural sweeteners					
Monk Fruit	57(62.0%)	10(10.9%)	8(8.7%)	13(14.1%)	4(4.3%)
Stevia	17(18.5%)	3(3.3%)	4(4.3%)	30(32.6%)	38(41.3%)
Xylitol	45(48.9%)	11(12.0%)	9(9.8%)	16(18.5%)	11(12.0%)

**Table 5 foods-11-01745-t005:** Consumption of low/zero-sugar products and zero-calorie sweeteners by consumer panel (at least once a month) (*n* = 92).

	Consumption Frequency, *n* (%)		
Variables	Yes	No	Don’t know
Low/zero-sugar foods/beverages	52(56.6%)	37(41.3%)	3(3.3%)
Artificial sweeteners			
Acesulfame-K	0(0.0%)	57(62.0%)	35(38.0%)
Aspartame	16(17.4%)	52(56.5%)	24(26.1%)
Erythritol	6(6.5%)	49(53.3%)	37(40.2%)
Saccharin	11(12.0%)	59(64.1%)	22(23.9%)
Sucralose	23(25.0%)	46(50.0%)	23(25.0%)
Natural sweeteners			
Monk Fruit	3(3.3%)	61(66.3%)	28(30.4%)
Stevia	26(28.3%)	43(46.7%)	23(25.0%)
Xylitol	11(12.0%)	48(52.2%)	33(35.9%)

**Table 6 foods-11-01745-t006:** The means ± standard error of overall liking, attribute liking, and purchase intent of sucrose, Reb A, D, and M ice cream samples (*n* = 92).

	Liking Score ^1^					Purchase Intent ^2,^***
Ice Cream	Overall *	Appearance *	Flavor ***	Texture/ Mouthfeel *	Aftertaste ***	
Sucrose	7.6 ± 0.13 ^a^	7.5 ± 0.12 ^a^	7.7 ± 0.12 ^a^	7.3 ± 0.16 ^a^	7.4 ± 0.13 ^a^	3.7± 0.12 ^a^
Reb A	5.4 ± 0.19 ^c^	6.7 ± 0.14 ^b^	5.2 ± 0.19 ^c^	6.1 ± 0.17 ^c^	4.3 ± 0.23 ^c^	2.1 ± 0.11 ^c^
Reb D	6.4 ± 0.16 ^b^	6.9 ± 0.17 ^ab^	6.2 ± 0.17 ^b^	6.4 ± 0.17 ^bc^	5.5 ± 0.19 ^b^	2.6 ± 0.11 ^b^
Reb M	6.6 ± 0.18 ^b^	7.1 ± 0.13 ^ab^	6.5 ± 0.19 ^b^	6.7 ± 0.14 ^ab^	5.6 ± 0.21 ^b^	2.8 ± 0.12 ^b^

^1^ The liking scores were evaluated on a nine-point hedonic scale (1 = Dislike extremely, 9 = Like extremely); ^2^ The purchase intent was evaluated on a five-point Likert scale (1 = Definitely would not buy, 5 = Definitely would buy); ^a,b,c^ values in the same column show the significant differences between sample means at *p* < 0.05 by Tukey’s (HSD). * indicates *p* < 0.05; *** indicates *p* < 0.001.

**Table 7 foods-11-01745-t007:** The means ± standard error for sweetness and bitterness intensities as rated by the consumer panel for sucrose, Reb A, D, and M ice cream samples (*n* = 92).

	Intensity ^1^	
Ice Cream	Sweetness ***	Bitterness *
Sucrose	10.3 ± 0.24 ^a^	1.6 ± 0.27 ^c^
Reb A	7.9 ± 0.40 ^b^	5.4 ± 0.37 ^a^
Reb D	8.0 ± 0.29 ^b^	2.9 ± 0.33 ^b^
Reb M	9.8 ± 0.30 ^a^	2.6 ± 0.36 ^bc^

^1^ Intensities measured immediately after swallowing on a 15-cm line scale (0 = Not at all sweet/bitter, 15 = Extremely sweet/bitter). ^a,b,c^ values in the same column show the significant differences between sample means at *p* < 0.05 by Tukey’s (HSD). * indicates *p* < 0.05; *** indicates *p* < 0.001.

**Table 8 foods-11-01745-t008:** Aftertaste attributes selected by the consumer panel for sucrose, Reb A, D, and M ice cream (*n* = 92).

	Ice Cream			
Attributes ^1^	Sucrose	Reb A	Reb D	Reb M
Artificial ***	7 ^a^	54 ^c^	47 ^bc^	35 ^b^
Bitter ***	2 ^a^	37 ^b^	14 ^a^	10 ^a^
Buttery ^ns^	22	18	17	27
Chemical ***	1 ^a^	26 ^c^	12 ^b^	10 ^ab^
Creamy ***	62 ^c^	32 ^a^	50 ^bc^	45 ^ab^
Honey ^ns^	5	4	6	4
Metallic **	2 ^a^	14 ^b^	5 ^ab^	8 ^ab^
Milky ***	62 ^b^	30 ^a^	49 ^b^	58 ^b^
Minty ^ns^	1	1	0	0
Pleasant ***	51 ^c^	9 ^a^	28 ^b^	29 ^b^
Sweet ***	72 ^c^	39 ^a^	53 ^ab^	62 ^bc^
Tart *	2 ^a^	13 ^b^	6 ^ab^	5 ^ab^
Vanilla ***	84 ^b^	54 ^a^	64 ^a^	64 ^a^

^1^ The listed terms for CATA analysis; ^a,b,c^ values in the same column show the significant differences between sample means at *p* < 0.05 by Critical Difference (Sheskin). * indicates *p* < 0.05; ** indicates *p* < 0.01; *** indicates *p* < 0.001 and ^ns^ indicates no significant differences among samples.

## Data Availability

No new data were created or analyzed in this study. Data sharing is not applicable to this article.

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
