# Peer review of "The Effect of Steviol Glycosides on Sensory Properties and Acceptability of Ice Cream"

_foods, 2022, doi:10.3390/foods11121745_

Round 1
Reviewer 1 Report
In this paper, the researchers investigated the sensory characteristics and acceptability of ice cream sweetened with Stevia A, D and M only using a consumer panel and compared stevia ice cream samples with sucrose ice cream. However, there are many issues with this article, which are commented on below.
- There are too many paragraphs in the introduction of the paper, and the focus of each paragraph is not clear, please divide it reasonably and revise it.
- There is a major problem with paragraph layout, so please check the layout carefully before and after submitting your manuscript so that it looks good and meets the requirements of this journal.
- There are also some problems with the formatting of tables in the article. Please mark the decimal point of Table 1 uniformly; Table 2 is disorganized; the last line of Table 4 is not aligned with the content of the previous line. Please check and revise them carefully. Line 193. Please adjust the table width so that the same content is in the same row. Line 128, 193, 200, 208, 215, 221, 251, 270. Please standardize the font and font size of the table content.
- The letters marked a,b,c for the significance of differences in Table 6,7,8 were not properly superscripted.
- The growth rate (CAGR) that appears for the first time in lines 34 and 64 of the article needs to be marked with the full name and abbreviation, and only the abbreviation is needed for the reappearance. As well as other locations in the article where such cases are found, it is recommended that a full-text check be performed and corrections made.
- There is a slight problem with the formatting of "kg/m2" in line 187, which does not need to be bolded, please make the change.
- The 56 lines of the paper: GI and its abbreviations should be positioned before and after the other full names and abbreviations in the text.
- “This study used 11 terms from a previous study conducted by Tao and Cho (2020) who evaluated aftertaste of stevia solutions” in line 159 is not properly labeled as a reference.
- Line 35, 49, 67, 76, 89, 97, 246, 334, 376, 396, 419, 426, 438. Please check the formatting.
- Line 221, 251, 270. Please use superscripts in the significance analysis.
- Line 35-66. It is suggested that these two paragraphs be streamlined and combined into one paragraph.
Author Response
Point 1: In this paper, the researchers investigated the sensory characteristics and acceptability of ice cream sweetened with Stevia A, D and M only using a consumer panel and compared stevia ice cream samples with sucrose ice cream. However, there are many issues with this article, which are commented on below.
response: The authors thank the reviewer for the valuable comments and appreciate elaborated critiques on this article. The following comments below have been changed/modified as suggested.
Point 2: There are too many paragraphs in the introduction of the paper, and the focus of each paragraph is not clear, please divide it reasonably and revise it.
Responses: The introduction has been revised and concise from 7 paragraphs to 4 paragraphs. The authors believe that each section of the revised introduction has been divided reasonably for the readers.
Point 3: There is a major problem with paragraph layout, so please check the layout carefully before and after submitting your manuscript so that it looks good and meets the requirements of this journal.
Responses: The authors used the format template provided by Foods Journal (docx.) including fonts, paragraph length and width, and line spacing.
Point 4: There are also some problems with the formatting of tables in the article. Please mark the decimal point of Table 1uniformly; Table 2 is disorganized; the last line of Table 4 is not aligned with the content of the previous line. Please check and revise them carefully. Line 193. Please adjust the table width so that the same content is in the same row. Line128, 193, 200, 208, 215, 221, 251, 270. Please standardize the font and font size of the table content.
Responses: The authors made a change in the format for Table 1 to be more concise and uniform with other tables. Table 2 was modified to make them more organized by adding a line in between variables to separate the context. Table 4 was adjusted to make it clearer and was reduced in size to align with the context. The font and font size of the table have been standardized accordingly to meet the requirements of this journal. Line 128, 193, 200, 208, 215, 221, 251 and 270 are now line 110, 175, 181, 189, 196, 202, 232, and 249. These were changed to have the correct font size and justified.
Point 5: The letters marked a,b,c for the significance of differences in Table 6,7,8 were not properly superscripted.
Responses: The authors changed a,b and c to be superscripted in Tables 6,7, and 8.
Point 6: The growth rate (CAGR) that appears for the first time in lines 34 and 64 of the article needs to be marked with the full name and abbreviation, and only the abbreviation is needed for the reappearance. As well as other locations in the article where such cases are found, it is recommended that a full-text check be performed, and corrections made.
Responses: All full names and abbreviations have been revised and corrected so that there will be no confusion for the readers. (Line 52)
Point 7: There is a slight problem with the formatting of "kg/m" in line187, which does not need to be bolded, please make the change.
Responses: Corrected; Line 187 is now line 168.
Point 8: The 56 lines of the paper: GI and its abbreviations should be positioned before and after the other full names and abbreviations in the text.
Responses: All full names and abbreviations have been revised and corrected. (Line 42)
Point 9: “This study used 11 terms from a previous study conducted by Tao and Cho (2020) who evaluated aftertaste of stevia solutions” in line 159 is not properly labeled as a reference.
Responses: Corrected; Line 159 is now line 140.
Point 10: Line 35, 49, 67, 76, 89, 97, 246, 334, 376, 396, 419, 426,438. Please check the formatting.
Responses: Corrected. The contexts in lines 35 and 49 were deleted. Line 67, 76, 89, 97, 246, 334, 376, 396, 419, 426 and 438 are now line 64, 72, 54, 58, 227, 310, 352, 373, 394, 400 and 413. These lines were justified to have the paragraphs adjusted to the same length.
Point 11: Line 221, 251, 270. Please use superscripts in the significance analysis.
Responses: Line 221, 251, and 270 are now 201, 232, and 249. All the significance analysis letters have been changed to superscripts so that the Tables look clearer to follow.
Point 12: Line 35-66. It is suggested that these two paragraphs be streamlined and combined into one paragraph.
Responses: The authors moved, adjusted, and combined paragraphs in introduction section.
Reviewer 2 Report
This article studied the effect of steviol glycosides (rebaudioside A, D, and M) on sensory properties and acceptability of ice cream. The experimental design (except the final concentration of the SGs in the ice cream) and results analysis to reach the goal are adequate.
However, the finding is expectable since this research is limited by the sensory features of the SGs, in another words, it is well known that rebaudioside D and M possess much better sensory properties than RA, almost in all application cases. Therefore, if the author can make some scientific explanation to this phenomena, by experiment or calculation, then this research could be completed.
My main concern is the concentration applied. 0.09% is 900ppm, which is fine for RD and RM, not for RA. RA of 900ppm is unbearable. The experiment should be based on same sweetness.
The title, suggest to delete “(rebaudioside A, D, and M)”.
Author Response
Point 1: This article studied the effect of steviol glycosides (rebaudioside A, D, and M) on sensory properties and acceptability of ice cream. The experimental design (except the final concentration of the SGs in the ice cream) and results analysis to reach the goal are adequate. However, the finding is expectable since this research is limited by the sensory features of the SGs, in another word, it is well known that rebaudioside D and M possess much better sensory properties than RA, almost in all applications cases. Therefore, if the author can make some scientific explanation to this phenomena, by experiment or calculation, then this research could be completed. My main concern is the concentration applied. 0.09% is 900ppm, which is fine for RD and RM, not for RA. RA of 900ppm is unbearable. The experiment should be based on same sweetness.
Response: The authors thank you for the comment referee provided. It has been well-known that Reb D and M provide better sensory properties than Reb A. However, our research objective was to see if Reb D and M could be used as a sole sweetener in high sugar applications such as ice cream (14% sucrose). In a previous study (Tao, R.; Cho, S. Consumer-Based Sensory Characterization of Steviol Glycosides (Rebaudioside A, D, and M). Foods 2020, 9, 572 1026, doi:10.3390/foods9081026), we first matched the sweetness level of Reb M to the 14% sucrose to use in high sugar applications such as ice cream, which was 0.09%. We then used the same concentration for Reb A and D so that we compare the sweetness and bitterness of Reb A, D, and M at the same concentration. We confirmed that Reb A of 900 ppm was unbearable, but Reb D and M were comparable to sucrose ice cream. We also found that Reb A had less sweetness than Reb D and M at the same concentration and thus we chose to use the same concentration of 0.09% Reb A, D, and M and compared to 14% sucrose. The authors also mentioned that this is continuous research in the manuscript (lines 98 – 100).The optimal ratio between Reb A, D, and M in order to find the best taste ratio in regard to both taste and product price will be investigated in future studies.
Point 2: The title, suggest to delete “(rebaudioside A, D, and M)”.
Response: As suggested, we deleted “(Rebaudioside A, D, and M)” in the title.
Reviewer 3 Report
The research deals with the potential of using the three steviol glycosides as sweeteners in vanilla ice cream. Researchers have employed means of sensory analysis with consumers to investigate sensory characteristics and acceptability the products.
I’d like to point out that the sensory panel is rather small for consumer studies. Authors have already addressed some limitations of the study in L438-447, where the size of the panel should be included as well. Basing on this you should be careful with generalization to a wider population.
Besides, I’d like to as authors if they have tested the low/no sugar ice cream purchase and preferences for ice cream sweetened with other sweeteners but sucrose (similarly as for gender)? Since the proportion of “I don’t know” replies is quite low, the Yes/No buyers create the majority groups.
Beside the remarks above, I have some comments to the manuscript bulleted below:
L8: please use more appropriate term instead of “zero-calorie”.
L28: please check the reference and revise the statement – the recommendation addresses added sugar and not added sucrose.
L104: … compromising eating quality. It is unclear; do you mean sensory quality?
L459-602: The last sentence of the conclusions seems out of scope of the research presented. Please revise.
Author Response
Point 1: I’d like to point out that the sensory panel is rather small for consumer studies. Authors have already addressed some limitations of the study in L438-447, where the size of the panel should be included as well. Based on this you should be careful with generalization to a wider population. Besides, I’d like to as authors if they have tested the low/no sugar ice cream purchase and preferences for ice cream sweetened with other sweeteners but sucrose (similarly as for gender)? Since the proportion of “I don’t know” replies is quite low, the Yes/No buyers create the majority groups.
Response: The authors thank the referee for the comments about this study. The authors attempted to do the test between the low/no sugar ice cream purchase and preferences for ice cream sweetened with other sweeteners as suggested. However, no significant difference was found (P = 0.790). The authors used two-way ANOVA with one interaction effect (fixed effects: ice cream sample and Yes/No buyers).
Beside the remarks above, I have some comments to the manuscript bulleted below:
Point 2: L8: please use more appropriate term instead of “zero-calorie”.
Response: The authors changed the term for stevia from “zero-calorie” to “a natural non-caloric sweetener” instead. (Line 8, 41 and 440)
Point3: L28: please check the reference and revise the statement – the recommendation addresses added sugar and not added sucrose.
Response: The authors want to use the term sucrose instead of sugar for the homogeneity of the article. However, the authors agree with the comment the reviewer made and thus we changed it to added sugar (Line 26).
Point 4: L104: … compromising eating quality. It is unclear; do you mean sensory quality?
Response: That is correct. The authors mean the sensory quality of consumption of food product with high-intensity sweeteners. However, since it’s unclear, the authors agreed to change from eating quality to sensory quality instead (Line 88).
Point 5: L459-602: The last sentence of the conclusions seems out of the scope of the research presented. Please revise.
Response: The future study directions were focused on the increase of the usage of minor steviol glycosides since there are several challenges of using these minor glycosides because of their small concentrations in the stevia leaves.
Last Sentence of Conclusion (Line 437- 442)
Further studies are needed to commercially use Reb D and M though since their small quantities in the stevia leaves. Breeding for increased concentrations of these minor glycosides in stevia plants would be beneficial to increase the supply of the desired glycosides. Also, identifying an optimal combination of different steviol glycosides to use both major and minor steviol glycosides to accommodate the bitter aftertaste issue from Reb A and small extractable quantities of Reb D and M.
Reviewer 4 Report
The manuscript "The Effect of Steviol Glycosides (Rebaudioside A, D, and M) on Sensory Properties and Acceptability of Ice Cream" investigated sensory characteristics and consumer's acceptability of ice cream sweetened solely with Reb A, D, and M steviol glycosides (compared to sucrose), using a consumer panel. I consider it is an useful study for the food industry and I positively appreciate this work, but the manuscript still can be improved and some suggestions are:
From the first glance, Table 1 is not clear enough. I suggest thorough explanation of its construction.
All abbreviations must be explained at first use. For example, line 186: BMI
The English language needs some slight improvement.
I noticed the repeated reference to a previous article of 2 of the authors (Tao, R.; Cho, S. Consumer-Based Sensory Characterization of Steviol Glycosides (Rebaudioside A, D, and M). Foods 2020, 9, 572 1026, doi:10.3390/foods9081026), which was referred 7 times in this manuscript. I noticed that the study concerns same steviol glycosides (Rebaudioside A, D, and M) and I understand the sense of its mention. But, still, I was wondering if any of these 7 times could be replaced with other authors reference.
Author Response
- The manuscript "The Effect of Steviol Glycosides (Rebaudioside A, D, and M) on Sensory Properties and Acceptability of Ice Cream" investigated sensory characteristics and consumer’s acceptability of ice cream sweetened solely with Reb A, D, andM steviol glycosides (compared to sucrose), using a consumer panel. I consider it is an useful study for the food industry and I positively appreciate this work, but the manuscript still can be improved and some suggestions are: From the first glance, Table 1 is not clear enough. I suggest thorough explanation of its construction.
Response: The authors thank the reviewer for the appreciation of the work. Table 1 has been modified to be more concise and clearer for the reader.
- All abbreviations must be explained at first use. For example,line 186: BMI
Responses: Line 186 is now line 168. The BMI and all other abbreviations have been revised and changed accordingly.
- The English language needs some slight improvement.
Responses: The authors worked with a writing expert who has extensive experience in writing scientific articles.
- I noticed the repeated reference to a previous article of 2 of the authors (Tao, R.; Cho, S. Consumer-Based SensoryCharacterization of Steviol Glycosides (Rebaudioside A, D, andM). Foods 2020, 9, 572 1026, doi:10.3390/foods9081026), which was referred 7 times in this manuscript. I noticed that the study concerns same steviol glycosides (Rebaudioside A, D, and M) and I understand the sense of its mention. But, still, I was wondering if any of these 7 times could be replaced with other authors’ reference.
Response: The reference was used as this paper was under the overarching goal of the same project to compare Reb A, D, and M in solution and food applications. Therefore, this experiment was necessary for a complete, coherent discussion. However, we agreed with the reviewer’s comment, and thus the authors attempted to find other articles that can replace the Tao and Cho (2020) reference.